# Mediterranean Diet and Thyroid: An Interesting Alliance

**DOI:** 10.3390/nu14194130

**Published:** 2022-10-04

**Authors:** Giuseppe Bellastella, Lorenzo Scappaticcio, Francesco Caiazzo, Maria Tomasuolo, Raffaela Carotenuto, Mariangela Caputo, Stefania Arena, Paola Caruso, Maria Ida Maiorino, Katherine Esposito

**Affiliations:** 1Department of Advanced Medical and Surgical Sciences, University of Campania “Luigi Vanvitelli”, 80138 Naples, Italy; 2Unit of Endocrinology and Metabolic Diseases, University Hospital, University of Campania “Luigi Vanvitelli”, 80138 Naples, Italy

**Keywords:** Mediterranean diet, thyroid function, thyroid cancer, thyroid autoimmunity

## Abstract

The Mediterranean diet, recognized as being cultural heritage by UNESCO, is mostly plant-based and includes a high consumption of whole-grain, fruit, and vegetables with a moderate consumption of alcohol during meals. Thus, it provides a small amount of saturated fatty acids and a high quantity of antioxidants and fiber. For this reason, it has been considered to have an important role in preventing cardiovascular diseases, chronic kidney diseases, type 2 diabetes mellitus, and cancer, but its relationship with thyroid function and diseases is still under debate. The aim of this review was to search for the possible correlation between the Mediterranean diet and thyroid function, and to critically evaluate the pathophysiological link between selected food intake and thyroid disorders.

## 1. Introduction

The traditional Mediterranean diet (MD) is part of the cultural inheritance of people living in the Mediterranean basin, and is defined as the consumption of mainly agricultural food. As a demonstration of worldwide importance, UNESCO (the United Nations Educational, Scientific, and Cultural Organization) has recognized it as a cultural heritage [1]. MD is characterized by a large consumption of fruit, vegetables, legumes, nuts, fish, complex carbohydrates, and extra virgin olive oil (EVOO), with a moderate consumption of alcohol (red wine) during meals and a low intake of processed meats, red meats, and sweets. The MD pattern provides a small amount of saturated fatty acid and a high quantity of antioxidants and fiber derived from fruits, vegetables, and EVOO. For all of these reasons, it has been included among the most beneficial dietary approaches to prevent major noncommunicable diseases, such as cardiovascular diseases (CVDs), type 2 diabetes mellitus (T2DM), chronic kidney diseases (CKDs), and cancer [2,3,4,5,6,7,8].

Various evidence has shown that metabolic diseases and thyroid dysfunctions are closely linked; in fact, the thyroid gland is the most important center of metabolic regulation.

The basal metabolic rate (BMR), oxygen consumption, and heat production depend on the release and activity of thyroid hormones. Free triiodothyronine and thyrotropin levels have been shown to be directly associated with abdominal adiposity, independent of insulin resistance [9].

Regarding diet, in the literature, soy and cruciferous vegetables have been described as being goitrogenic foods, because they are rich in goitrin, which acts as a disruptor of thyroid function through the inhibition of thyroid hormone synthesis; even iodine-rich food, such as cereals, seafood, beef, poultry, and milk, could led to thyroid disturbances, in cases of both low and excessive intake [10,11,12]. On the other hand, selenium and zinc have been described as beneficial oligo-elements for thyroid function, preventing cell damage and premature aging [13,14]. Vitamin D and calcium also play a role in thyroid function and diseases [15].

As MD presumes the intake of a large amount of food rich in various oligo-elements, it could be useful to know the impact of this dietary regimen on thyroid function; however, surprisingly, there are only a few studies that have investigated the possible influence of MD on thyroid physiology and pathology.

In this regard, there are some useful recommendations for patients regarding their diet, such as the regular consumption of vitamin D (fatty fish and fish oil), B group vitamins (meat, wholegrain cereals, and chicken eggs), vitamin A (carrot, pumpkin, and spinach), vitamin C (vegetables and fruits), magnesium (yogurt and nuts), zinc (meat, mushrooms, and wholegrain cereals), iron (meat, spinach, and seafood), iodine (seafood and fish), and selenium (walnut and fish) [16,17]. A portion of the protein, vitamins, and mineral salts can be consumed in the form of legumes (beans, chickpeas, broad beans, peas, and lentils), especially for vegetarians and vegans [18]. Among legumes, even soy foods, known to have an inhibiting effect on thyroperoxidase, do not seem to cause a significant disruption in thyroid function [19].

However, it has to be considered that some foods may be polluted by chemical substances, known as endocrine disruptors (EDs), which show a strong interference with thyroid gland function at various levels, impairing hormone synthesis, metabolism, and bioavailability [20,21,22,23,24].

One of the most common EDs that may impair thyroid function is Bisphenol A (BPA), an organic chemical compound that belongs to the group of phenols used in the production of plastics, including polycarbonates and epoxy resins. The presence of bisphenols has been demonstrated in various food, mostly in packaged foods and drinks [25]. High concentrations have been found in seafood, canned fish, and canned vegetables. BPA can affect thyroid function in various ways, impairing pituitary and thyroid gene expression, inducing cellular toxicity and, in addition, have an antagonistic effect on thyroid receptors (TRs) [26].

A study by Berto-Júnio et al. investigated the interaction between BPA and PAX8 (Paired Box Protein 8) and TTF1 (Thyroid Transcription Factor 1), which play important roles in thyroid organogenesis and hormone production [27].

Similar to BPA, phthalates (PAE) act as EDs. A review conducted by Giuliani et al. analyzed articles on the presence of PAE in food: oils, meat, dairy products, and plants; the most contaminated were commercial wines, both white and red, and because of its affinity for fat, oils and fat-rich meats and milk [28].

Usually, products that are part of MD are grown following the criteria of organic farming, and are thus less polluted by endocrine disruptors [29,30]. So, the negative effects on thyroid function are negligible.

The aim of this review was to search for the possible correlation between MD and thyroid function, and to critically evaluate the pathophysiological link between selected food intake and thyroid disorders.

## 2. Materials and Methods

This review was aimed at highlighting the main results from studies on this issue that appeared in the literature until 31 May 2022. For this purpose, a systematic Pubmed and Scopus engine search was carried using a set of keywords, as follows: “mediterranean diet” and “thyroid cancer”, “hashimoto thyroiditis”, “thyroid autoimmunity”, “hypothyroidism”, and “hyperthyroidism”. Reviews, metanalyses, original papers, and observational studies reporting results on the correlation between the Mediterranean diet or its components and thyroid disturbance in term of cancer, autoimmunity, hypothyroidism, and hyperthyroidism were included and the contents were critically evaluated.

## 3. Epidemiology of Thyroid Diseases

Iodine plays a pivotal role in thyroid physiology and pathology. The ideal daily intake of iodine recommended by the World Health Organization (WHO) is 150 μg for adults, with an increase to 250 μg in pregnant and lactating women. A population is considered by WHO to be iodine-deficient when its median urinary excretion is <100 μg /L. Unfortunately, despite political efforts to ensure an adequate iodine intake by introducing the use of iodinated salt, several industrialized countries, previously considered as iodine-sufficient, still show iodine deficiency as an emerging issue [31].

Simple goiter without thyroid dysfunction is the most common thyroid disease, and is strictly correlated to iodine deficiency. It is classified as endemic goiter when it affects more than 5% of preadolescent school-age children. The ratio of women to men is 4:1, with the higher prevalence of diffuse goiter in premenopausal women. In fact, the mean frequency is about 26% (31% in those aged <45 years and 12% in those aged >75 years, which show prevalently a nodular goiter). The frequency of thyroid nodules is estimated to be 5% in women and 1% in men, and ultrasonography imaging is thought to be the most appropriate tool to discover thyroid nodules [31,32].

Congenital hypothyroidism is found in 3 newborns per 10,500–12,000 births, and is usually associated with treatable mental retardation [31]. Acquired primary hypothyroidism is usually associated with autoimmune thyroiditis or is the consequence of radioiodine treatment for hyperthyroidism or thyroidectomy for goiter, cancer, or relapse of Graves’ disease after treatment with anti-thyroid drugs. Hypothyroidism may also occur in patients treated with amiodarone, lithium, interferon, or therapy with check-point inhibitors. The prevalence of hypothyroidism in iodine-replete populations ranges from 1.2 to 4.0 per 1000 in men and 0.6 to 12 per 1000 in women, with differences among these ranges in the USA, Japan, and Europe. Curiously, areas of iodine deficiency show a lower prevalence [31]. Subclinical hypothyroidism (increased TSH levels with normal FT3 and FT4 levels, usually associated with autoimmune thyroiditis in asymptomatic patients, occurs in 3% of men and 10% of women according to the Whickham Survey, whereas high TSH levels have been found in 9.4% of subjects according to the Colorado study [33,34].

Hyperthyroidism is most frequently associated with Graves’ disease and less frequently with toxic adenoma and toxic multinodular goiter, and more rarely with thyroiditis or drugs (iodine or amiodarone). Its prevalence ranges from 0.5 to 2%, with a ratio of 10/1 of women to men [33]. Subclinical hyperthyroidism (low TSH levels with FT3 and FT4 still normal) has a prevalence of about 3%, which is higher before 40 years old and from 80 years old onwards [35]. Thyroid cancer is the most common endocrine cancer, which has recently been showing an increase in prevalence and incidence, even if its mortality has remained low. The cancers originate from the epithelial cells, and are grouped as follows:-differentiated thyroid cancer, which includes papillary, follicular, and Hurthle cell cancer;-poorly differentiated thyroid cancer and anaplastic thyroid cancer.

Instead, medullary thyroid cancer originates from neuroendocrine C cells.

According to worldwide Globocan 2020 statistics, thyroid cancer reached an incidence of 6.6/100,000 with a mortality rate of 0.43/100,000. This cancer has been found to be more common in Asia, in women, and in those 35–64 years old. In addition, papillary thyroid cancer was found to be the most common histological subtype, accounting for 90% of new cases, and it has the best prognosis [36], even if most of the known risk factors for the development of thyroid cancer are family history, female sex, exposure to ionizing radiation in the head and neck, advanced age, iodine deficiency or excess, and tobacco and alcohol use. Several metabolic disorders, obesity, insulin resistance, and diabetes mellitus have recently been associated with a higher incidence of thyroid carcinoma [37,38]. The increasing evidence of the association of thyroid disease, including cancer, with metabolic disorders, and the different prevalence of these diseases in countries with different eating habits are attracting the attention of researchers regarding the possible preventative role of MD for the development of these diseases.

## 4. Mediterranean Diet and Thyroid Autoimmunity

Hashimoto disease (HT) and Graves’ disease (GD) are examples of autoimmune thyroid disease (AITD).

HT is one of the most common thyroid diseases, and is characterized by lymphocytic infiltration of the thyroid tissue and the presence of antibodies of thyroid peroxidase and thyroglobulin. The disease is more frequent in women than in men, and in those aged 30–60, although it can be diagnosed in patients at any age. AITD development is the result of a combination of genetic factors (major histocompatibility complex, immune-modulating genes, and thyroid specific genes), and existential (parenthood, age, and female gender), environmental (smoking, exposure to heavy metal, and endocrine disruptors (ED)), and nutritional (iodine excess or deficiency, selenium deficiency, alcohol consumption, and vitamin D intake) factors.

Food is considered to be an important environmental factor that plays a role in the development of AITD. A study by Tomer et al. concluded that, although the pathogenetic mechanism is still unknown, several environmental factors, such as dietary ones (perhaps micronutrients or the gut microbiota), are involved in the development of HT and autoimmunity overall, in genetically susceptible patients [39].

In a recent study by Ruggeri et al., a total of 200 volunteers were enrolled, 81 of which had been diagnosed as being affected by HT by currently accepted laboratory and ultrasonographic criteria (serum antithyroid antibody positivity and/or heterogeneous echo-structure with diffuse or patchy hypo-echogenicity at ultrasound), and the remaining 119 served as the controls [40]. Each volunteer underwent a physical examination, clinical history, and thyroid ultrasonography; moreover, a PREDIMED questionnaire was administered to ascertain MD adherence. HT patients showed a higher consumption of animal foods (especially red meat and processed food) in contrast with the controls, who showed a high intake of plant foods, including legumes and fresh fruits and vegetables. The authors also found a statistically lower consumption of nuts in HT patients than in the controls (*p* = 0.0005). In this pilot study, HT patients showed a high intake of animal proteins, saturated fats, and refined sugar—in other words the prototype of the so called Western-type diet, in contrast with the Mediterranean one.

A 2020 large cohort study by Kaličanin et al. aimed to identify the different consumption of food groups and the frequency intake between HT patients and controls by using a food frequency questionnaire (FFQ) [41]. The FFQ consisted of 51 items concerning foods and beverages. The frequency of intake of each food item was measured using six categories: every day, 2–3 times a week, once a month, once a week, rarely, and never. Dietary intake in the control participants was assessed using an FFQ consisting of 54 items regarding foods and beverages. The frequency of food intake was measured using five categories, as in the HT patients. Additionally, there was a question in both FFQs regarding fat consumption, with three choices (plant oil, olive oil, and animal fat) and three frequency categories (always, sometimes, and never). The authors grouped 48 food items that were common in both questionnaires into 22 food groups, and they converted the frequency categories into weekly intake for each of the 48 food items.

The HT patients showed a significantly increased consumption of animal fat and processed meat, whereas the controls consumed red meat, non-alcoholic beverages, whole grains, and plant oil significantly more frequently. Moreover, the authors found a significant positive association between plant oil consumption and triiodothyronine levels in HT patients, and a negative association between olive oil consumption and systolic blood pressure in the sub-group of HT patients on levothyroxine therapy [41].

It is known that the triglycerides in animal fats contain saturated fat acids (SFA) and monounsaturated fat acids (MUFA). SFAs are associated with the development and progression of various chronic diseases through inflammatory responses [42]. Two studies have shown that a high-fat diet causes thyroid dysfunction in rats, which led to hypothyroidism by decreasing the T4 levels [43,44]. These results are in line with a study by Matana et al., who showed a significantly (*p* = 0.01) higher frequency in the positive plasma TPO and Tg antibodies in the group with an increased intake of animal fats and butter [45].

In particular, processed meat, such as bacon, sausages, and salami, which contain mostly fats, few proteins, and very low carbohydrates, represents another food group highly consumed in HT patients. Moreover, processed meat could contain nitrate and nitrite, with sodium used as additives.

A recent review showed that a high exposure to dietary nitrite or nitrate led to histological changes in the thyroid tissue and to a decrease in the plasma levels of thyroid hormones, by binding sodium-iodide symporter, and thus inhibiting iodine uptake [46,47,48].

In contrast, food groups less used among HT patients were as follows:-Red meats, which are good sources of zinc, iron, and selenium, which are essential nutrients for normal thyroid function. Red meat is rich in vitamin B12, whose deficiency is associated with autoimmune thyroid disease. The difference between fresh and processed red meats is the low fat content and the absence of additives. Moreover, processed meats are classified as a Class 1 carcinogen by the World Health Organization.-Non-alcoholic beverages such as multivitamin fruit juice. This report is in line with the literature, where there is a correlation between a scarce intake of vitamins and thyroid disease. In this regard, a recent study showed decreased levels of anti-thyroid antibodies in patients treated with vitamin C [49].-Whole grains rich in fiber, which is energy for the gut microbiota that produce short-chain fatty acids (SCFAs), which are important for cell proliferation and immune system function. A recent study found that muesli consumption was associated with a low risk of positive antibodies [45].-Plant oils, such as pumpkin seed oil, sunflower oil, and olive oil, which are rich in polyunsaturated fatty acids (PUFAs) and have anti-inflammatory properties. Olive oil in particular has been found to be associated with a decreased risk of autoimmune disorders such as lupus erythematosus and rheumatoid arthritis [50,51]. Oleocanthal, contained in extra virgin cold pressed olive oil, has an ibuprofen-like activity [52], which can explain the reported anti-inflammatory and immunomodulatory effects found from the regular consumption of olive oil. A systematic review by Pang et al. showed a thyroid-protective effect of EVOO in animal models. These mechanisms are unknown, but olive oil, olive leaf extract, and olive pomace residues seem to stimulate thyroid function in euthyroid and hypothyroid animals [53].

A moderate consumption of alcohol, as suggested in MD, is associated with a reduced risk of HT [54].

Oily fish, eicosapentaenoic acid (EPA), and docosahexaenoic acid (DHA) may reduce inflammation; a recent study showed an inverse correlation between the intake of oily fish and serum thyroid antibodies during pregnancy [55]. Moreover, oily fish and seafood are rich in selenium, iodine, iron, and zinc, which have a beneficial role in autoimmune thyroid disease.

Fruits, beside the already-mentioned beneficial roles of vitamins, minerals, and dietary fiber, additionally contain phytochemicals such as polyphenolic compounds that are known for their anti-inflammatory and antioxidant effects on human health [56].

A nutritional factor influencing thyroid function includes micronutrients, whose levels have been found to be impaired in AITDs.

Iodine is an important component of MD, and is contained in seafood; milk; eggs; fish as cod, haddock, and scampi; and in mineral water, as well as in iodine salt. It is well established that iodine is the most important oligoelement for thyroid function; a low intake can cause goiter, but a high intake (>1 mg/daily) may lead to reduced thyroid function and to the Wolff–Chaikoff effect [57]. An excess of iodine is toxic to cells; it can initiate apoptosis or necrosis. Some in vitro studies on the cells of patients with Hashimoto’s disease have shown that an excessive amount of iodine stimulates apoptotic processes and increases the production of free radicals [58]. A low iodine intake can lead to nodular goiter, with a rise in Abs levels due to antigens released from the abnormal gland. Its excess, especially in sufficient iodine areas, and its increased intake from deficiency, is associated with thyroid dysfunction and with autoimmunity. In a Chinese study, a higher prevalence of Abs titer (2.8%) was found in areas of excessive iodine intake [59].

The pathogenic mechanisms through which increased iodine intake induces autoimmune disease are still unclear. The hypothesis could be that high iodine exposure, in genetically susceptible individuals, may increase the immunogenicity of thyroglobulin, induce the auto-antigen presentation activity of thyrocytes and dendritic cells, impair peripheral tolerance by inhibiting regulatory T cells, cause oxidative stress leading to thyroid tissue injury, activate auto-reactive T cells that increase cytokine secretion, and eventually trigger apoptosis-signaling pathways, leading to thyroid destruction [60].

Selenium is contained in yeasts and in everyday products. Along with proteins, such as meats and fish, unprocessed cereals are also rich in selenium. Selenium is protective against autoimmune thyroid disease, and a deficiency in intake can affect the thyroid, which is susceptible to the negative effects of excess iodine and inhibits enzymes with selenocysteine residues, such as glutathione, thus inhibiting the action of antioxidant enzymes. This process may be reversible when given selenium [61]. A recent study conducted among people with similar genetic, environmental, and lifestyle factors and a comparable iodine status, but a very different selenium state, showed that the prevalence of thyroid dysfunction as lower in the adequate-selenium population than in the low-selenium population [62]. A recent meta-analysis of 16 trials showed the protective effects of selenium supplementation in lowering serum thyroperoxidase antibodies levels after three months and thyroglobulin antibodies titers after twelve months [63]. Despite missing randomized trials comparing selenium supplementation and HT, a beneficial effect of selenium on autoimmune thyroid diseases is reasonable. Selenium, as an antioxidant, may have a protective function [64], it can up-regulate regulatory T-cells, resulting in increased immune tolerance [65]. Moreover, selenium has anti-inflammatory effects [66], and it may suppress the expression of HLA-DR molecules on thyrocytes, reducing the development of thyroid autoimmunity [67].

Zinc is contained in seeds, such as flax seeds and pumpkin seeds, and in whole-grain cereals, such as buckwheat, millet, and whole-meal bread. Zinc is important for the production of thyroid hormones, and its scarce intake leads to an increase in antibody titers against thyroid antigens. Clinically, zinc deficiency in hypothyroidism could lead to hair loss [68].

Iron is part of the thyroid peroxidase (TPO) enzyme, which is required for hormone synthesis. Iron deficiency may contribute to lowering thyroid hormone levels, with a consequent increase in TSH and enlargement of the thyroid gland [10]. A high iron concentration can be found in meat, fish, and dark green vegetables.

Vitamin D status may be involved in the development of autoimmunity, related to its role in inducing excessive activation of Th1 and Th17 cells, as well as in impairing the function of regulatory T-cells and causing a deficiency in CD8+ T-cells, which is implicated in the pathogenesis of autoimmune thyroid disturbance (AITD) [69]. Some studies have reported lower levels of 25OH Vitamin D in HT patients than in the healthy controls. Furthermore, evidence has shown that for every 5 nmol/L increase in plasma 25OH Vitamin D levels, a 1.67 times lower HT risk is found. Moreover, an inverse relationship between 25OH vitamin D levels and thyroid antibodies titers in HT patients has been seen in a lot of studies [70,71,72,73,74,75,76,77,78,79].

## 5. Mediterranean Diet and Thyroid Cancer

As described in the paragraph regarding the epidemiology, the incidence rates of thyroid carcinoma (TC) have risen rapidly in high-income countries, and the disease is more common among women [80,81]. Differentiated thyroid cancers, such as papillary and follicular carcinoma, represent 98% of thyroid cancer [81,82]. The only definite risk factors for thyroid carcinoma are exposure to ionizing radiation, especially in childhood [83]; thyroid adenoma; and a history of goiter [84,85].

Moderate alcohol consumption is a part of MD, and for this reason, we searched in the literature for a correlation between alcohol and thyroid cancer. In a multicenter prospective study, Abhijit Sen et al. analyzed a cohort of 477,263 participants and 556 incident-differentiated thyroid carcinoma cases [86]. It was observed that moderate alcohol intake at recruitment was associated with a statistically significant lower risk of thyroid carcinoma. Compared with participants consuming 0.1–4.9 g of alcohol per day at recruitment, participants consuming 15 g or more (approximately 1–1.5 drinks) had a 23% lower risk of thyroid carcinoma (HR = 0.77; 95% CI = 0.60–0.98) and did not show differences when considering baseline or lifetime alcohol consumption, alcoholic beverage type, thyroid carcinoma histology and stage, age, sex, BMI, smoking status, and diabetes. Moreover, for every 10 g of alcohol consumed per day among consumers, the risk of thyroid carcinoma was lowered by 9% (HR = 0.91; 95% CI = 0.84–0.98) [86].

Large cohort and case-control studies have described inverse associations between moderate alcohol consumption and risk of thyroid carcinoma [87,88,89,90,91,92,93,94]. The Women’s Health Initiative Cohort Study, which included 159,340 post-menopausal women with 331 incident thyroid cancer cases, reported a non-significantly lower risk of thyroid cancer in consumers of ≥7 drinks per week vs. none (HR = 0.66; 95% CI = 0.44–1.01) [91].

In this regard, a pooled analysis of five prospective studies from the United States, which included 384,443 men and 361,664 women, and 1003 incident thyroid cancers, showed a HR of 0.72 (95% CI = 0.58–0.90) for an alcohol intake of ≥7 drinks per week vs. zero [92].

Some studies have demonstrated a toxic effect from high alcohol consumption on the hypothalamus–pituitary–thyroid axis [95,96].

The pillar of MD is certainly fruit and vegetable intake. It is well known that the consumption of vegetables and fruits protects against cardiovascular disease and probably against lower-respiratory and digestive cancer risks. A multicenter prospective cohort study of WHO analyzing a cohort of over half a million participants over a follow up of 14 years identified 748 incidents of first primary differentiated thyroid cancer cases. In particular, this study evaluated the association between fruits, vegetables, and fruits juice intake consumption and the differentiated thyroid cancer risk [97]. They found no association between the consumption of fruits and all differentiated (papillary or follicular) thyroid cancer. Instead, a positive marginal trend between fruit juice consumption and the total risk for differentiated thyroid cancer was detected, probably due to the high sugar content. A meta-analysis of 19 case-control studies and a pooled analysis of 11 case-control studies also found a correlation between vegetable intake and TC risk, showing a weak inverse association with the intake of total vegetables, excluding cruciferous ones [98,99]. In this regard, the results of an American cohort study of 300,000 participants suggested that a diet rich in cruciferous vegetables such as broccoli and cauliflower during early adulthood could have an impact on the early development of TC [100]. A pooled analysis of four Italian-based case-control studies demonstrated an inverse association among citrus fruits and TC risk, because of their high content of antioxidants (such as vitamin C and flavonoids), polyphenols, and fiber [101,102,103,104]. However, fruit juice, if polluted by traces of chemical pesticides, may be associated with a higher incidence of TC [105]. Even vegetables, because of their content of nitrates, have been shown to be positively associated with TC risk, because of their role in promoting carcinogens [106,107].

On the other hand, the results of a Korean case-control study, based on the comparison of a large cohort of women with malignant (TC) and benign (nodular goiter) thyroid diseases with a respective control group, demonstrated that the consumption of raw vegetables, persimmons, and tangerines had a significant protective effect against thyroid cancer [108].

In vitro studies have demonstrated that exposure to curcumin was able to induce a reduction in cell viability and an increase in apoptosis, both in anaplastic thyroid carcinoma-derived cell lines and in papillary thyroid cancer cells [109,110].

In addition, resveratrol, another particular nutraceutical, was able to induce apoptosis and cell cycle arrest, as well as antiproliferation and re-differentiation effects, in thyroid cancer cell lines in papillary and follicular thyroid cancer models [111,112]. It prevents oxidative injury caused by the radiolysis of water in thyroid tissues and inhibits DNA impairment during radioiodine therapy in anaplastic thyroid cancer models [113,114].

Moreover, the intragastric and intraperitoneal injection of resveratrol in rat models efficiently reduced the frequency and severity of DEN/MNU/DHPN-thyroid cancer-related lesions [112].

Flavonoids are plant pigments whose chemical structure is derived from that of the flavone. They are widely found in the plant world and include anthocyanins, flavones, and other pigments, and they have pharmacological properties, such as antioxidant and anti-inflammatory effects [115]. Recent in vitro studies have shown that some flavonoids can be beneficial for thyroid cancer, by reducing cell proliferation and increasing cell death [116,117]. Flavonoids have been shown to have antiproliferative and cell re-differentiation effects [118]. In particular, apigenin and luteolin are the most potent inhibitors of human thyroid carcinoma (papillary, follicular, and anaplastic carcinoma) cell lines in vitro, also inducing the re-expression of NIS mRNA in anaplastic thyroid carcinoma cell lines [117]. Moreover, the routine administration of flavonoid for 5 days seems to increase not only the thyroid radioiodine uptake, but also NIS protein and mRNA levels in the thyroids of animals [119]. Finally, a large cohort study including 748 patients with primary differentiated TC evaluated the effects of the intake of lean fish, fatty fish, fish products, and shellfish during a mean follow-up of 14 years [120]. The results demonstrated that there was no significant relationship between total fish consumption and risk of differentiated thyroid cancer.

## 6. Conclusions

Even if further studies on larger populations from different countries are needed to better clarify the relationships between the type of diet and thyroid disease, MD seems to have a beneficial effect on preventing these diseases, including cancer. However, it has to be taken into account that even the foods that characterize this diet may be contaminated by endocrine disruptors; thus, particular attention must be paid to the choice of foods and their origin from sustainable farming methods that ban or minimize the use of pesticides and other substances that can play the role of endocrine disruptors.

## Data Availability

Not applicable.

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
