# Peer review of "Mediterranean Diet and Thyroid: An Interesting Alliance"

_nutrients, 2022, doi:10.3390/nu14194130_

Round 1

Reviewer 1 Report

Dear Authors,

All diseases of the thyroid gland - tumors, goiter, hypothyroidism, hyperthyroidism, autoimmune and thyroid disturbance, are very common today, they are the scourge of humanity. It is no surprise that researchers all over the world are trying to find answers in an area that we have an influence on every day. Namely in the food you eat everyday. In addition, the Mediterranean diet is considered to be one of the healthier in the world. Especially since there aren't many studies on the effects of the Mediterranean diet on the thyroid gland.

The work reads very well. It is clear, understandable and has the correct layout. The aim of the work was clearly formulated, the method was chosen correctly. The literature is up-to-date and correctly cited.

The work thoroughly reviews the literature to explain what science knows today, the impact of diet on thyroid diseases. Analyzes the influence of particular groups of foods, spices and other consumed substances. The impact of food decontamination in the form of the so-called endocrine disruptors. It is valuable that studies from around the world are presented. The literature results are sometimes inconsistent and unambiguous, but undoubtedly most of the analyzes indicate a positive effect of MD on the health of the thyroid gland, with the note that food should come from crops free of contaminants and pesticides that interfere with the functioning of the thyroid gland.

The work is written in a simpe but interesting way and no major errors were found. There are two comments:

1. Little has been mentioned about the influence of legumes and their lectins on the functioning of the thyroid gland. Such an analysis will make the work complete, especially as there are more and more vegetarians worldwide for whom legumes are their primary source of protein. Won't the thyroid suffer?

2. The abbreviation “endocrine disruptor” should be harmonized - line 58 (translation of the abbreviation ED) should be aligned with line 72 (where the abbreviation is different - EDC)

Author Response

We are very grateful to this Reviewer for his pertinent comments and suggestions:

  1. Little has been mentioned about the influence of legumes and their lectins on the functioning of the thyroid gland. Such an analysis will make the work complete, especially as there are more and more vegetarians worldwide for whom legumes are their primary source of protein. Won't the thyroid suffer?

Following the suggestion by the Reviewer, the following sentences have been added in the text and related references have been added in bibliography: “a portion of protein, vitamins and mineral salts can be taken with legumes (beans, chickpeas, broad beans, peas, lentils) especially for vegetarians and vegans, (Dhaliwal SK, Talukdar A, Gautam A et al. Developments and prospects in imperative underexploited vegetable legumes breeding: A review. Int J Mol Sci 2020 21(24). 9615; doi: 10.3390/IJMS21249615).

Among the legumes, even soy foods, known to have an inhibiting effect on thyroperoxidase, do not seem to cause a significant disruption of thyroid function (Rizzo G, Baroni L. Soy, soy foods and the role in vegetarian diet. Nutrients 2018, 10(1), 43; doi: 10.3390/nur10010043)

  1. The abbreviation “endocrine disruptor” should be harmonized - line 58 (translation of the abbreviation ED) should be aligned with line 72 (where the abbreviation is different - EDC)

The abbreviation “endocrine disruptor” has been harmonized : ED both in line 58 and in line 72.

Reviewer 2 Report

Thank you for the invitation to review this interesting article.

The beneficial effect of the Mediterranean diet on metabolic disturbances (dyslipidemia, glucose alterations, diabetes, cardiovascular diseases) is well documented/known.

The goal of the manuscript is attractive - to verify the effect of this diet on thyroid function in (physiology and pathology).

However, I have some comments/concerns.

The most serious concern relates to the ‘materials and methods section. In my opinion, it is incomplete and needs a lot of further work. Chapter is missing:

§  The strategy for search and selecting studies (date range, specification of the inclusion and exclusion criteria for the review, how/if studies were grouped for the conclusion)

§  What was the number of studies screened, assessed for eligibility, and included in the review, with reasons for exclusions at each stage

PRISMA guidelines is worth reading.

Secondly, I think that a few sentences need to be re-edited:

Line 51-52 ‘At this regard, there are some useful recommendations for patients regarding the diet, such as regular consumption of vitamin D (fatty fish, fish oil) (…)

-          Whether we are able to provide the recommended dose of vitamin D with our diet?

Line 76-77’ Usually, products that are part of the MD are grown following the criteria of organic  farming and are therefore less polluted by endocrine disruptors. So that the negative effects on thyroid function are negligible’

-          Is this a personal opinion or a fact? it is worth giving the sources of such research. What about the fish and possible contamination in that case?

The chapter ‘Epidemiology of thyroid disease’ does not fully correspond to the topic of the article (In my opinion).

The cited study probably needs more detail - what are the differences in product consumption patterns: ‘A 2020 large cohort study by Kaličanin et al. aimed to identify different consumption of food groups and frequency intake between HT patients and controls by using 173 food frequency questionnaire (FFQ)’ line (172-174).

Line 193-194 ‘The difference between red fresh meat and processed ones is low fats content and absence of additives’.

-          is that the only difference? what about the fact that processed meat is classified as a Class 1 carcinogen by the World Health Organization.

In addition, among the search terms, the authors listed the Mediterranean diet (Materials and Methods), but most of the studies described are only nutrients/product groups whose impact is well known.

No studies in which the effect of the Mediterranean diet (as a pattern/per se) on the thyroid axis/thyroid hormone  metabolism was verified. Authors could have at least presented the amounts of the ingredients that were described (I, Se, Fe..) in the Mediterranean diet.

Moreover, is curcumin a characteristic part of the Mediterranean diet? (Line 334-336).

Discussion section is disproportionate to the rest of the text

Author Response

We are indebted to this Reviewer for his pertinent comments and criticisms. We are pleased to reply.

The most serious concern relates to the ‘materials and methods section. In my opinion, it is incomplete and needs a lot of further work. Chapter is missing:

      The strategy for search and selecting studies (date range, specification of the inclusion and exclusion criteria for the review, how/if studies were grouped for the conclusion)

      What was the number of studies screened, assessed for eligibility, and included in the review, with reasons for exclusions at each stage

PRISMA guidelines is worth reading.

We agree with the Reviewer that the Preferred Reporting Items for Systematic reviews and Meta-Analysis for Protocols (PRISMA) with its 27-item checklist is worth reading. However we think the method used to consider the studies on this issue so far appeared in the literature may be considered adequate. Anyway the related paragraph has been modified as follows:

This review was aimed at highlighting the main results from studies appeared in the literature until May 31th, 2022 on this issue. To this purpose, a systematic Pubmed and Scopus engine search was carried by using a set of keywords as follows: “mediterranean diet” and “thyroid cancer,” “hashimoto thyroiditis”, “thyroid autoimmunity”, “hypothyroidism”, “hyperthyroidism”. Reviews, meta-analyses, original papers, experimental and observational studies reporting results on correlation between Mediterranean diet or its components and thyroid disturbance in term of cancer, autoimmunity, hypothyroidism, hyperthyroidism were included and the contents critically evaluated.

Secondly, I think that a few sentences need to be re-edited:

Line 51-52 ‘At this regard, there are some useful recommendations for patients regarding the diet, such as regular consumption of vitamin D (fatty fish, fish oil) (…)’

      Whether we are able to provide the recommended dose of vitamin D with our diet?

The need for vitamin D varies with age and it is difficult to establish, for example, how much milk, fish, cereals must be consumed to ensure the right amount of vitamin D with the food. No data about this are reported in the literature.

Line 76-77’ Usually, products that are part of the MD are grown following the criteria of organic farming and are therefore less polluted by endocrine disruptors. So that the negative effects on thyroid function are negligible’

      Is this a personal opinion or a fact? it is worth giving the sources of such research. What about the fish and possible contamination in that case?

In the revised version two citations have been added after disruptors and reported in bibliography as follows:

- Mie  et al. A, Andersen HR, Gunnarsson S, et al. Human health implications of organic food and organic agriculture: a comprensive review. Environ Health 2017; 16(1):111; doi: 10.1186/s12940-017-0315-4

- Hurtado-Barroso S, Tressera-Rimbau A, Valleverdù-Queralt A, Lamuela-Raventos RM. Organic food and impact on human health. Crit Rev Food Sci Nutr 2019; 59(4): 704-714

The chapter ‘Epidemiology of thyroid disease’ does not fully correspond to the topic of the article (In my opinion).

We have great respect for the reviewer’s opinion but we believe that in a review on the relationship between the Mediterranean diet and thyroid physiopathology, a paragraph on the epidemiology of thyroid diseases may be useful for the reader.

The cited study probably needs more detail - what are the differences in product consumption patterns: ‘A 2020 large cohort study by Kaličanin et al. aimed to identify different consumption of food groups and frequency intake between HT patients and controls by using 173 food frequency questionnaire (FFQ)’ line (172-174).

Following the suggestion of this Reviewer, in the revised paper the following details  have been added:

The FFQ is the most commonly used dietary assessment tool for the evaluation of food consumption and measurement of long-term food intake. It consisted of 51 items concerning foods and beverages. The frequency of intake of each food item was measured using six categories: every day, 2–3 times a week, once a month, once a week, rarely, and never.   Dietary intake in control participants was assessed using the FFQ  consisting of 54 items regarding foods and beverages. The frequency of food intake was measured using five categories as in HT patients. Additionally, there was a question in both FFQs regarding fat consumption with three choices (plant oil, olive oil, and animal fat) and three frequency categories (always, sometimes, and never). The authors grouped 48 food items that were common in both questionnaires into 22 food groups and they converted the frequency categories into weekly intake for each of the 48 food items .  HT patient showed significantly increased consumption of animal fat and processed meat , whereas controls consumed significantly more frequently red meat , non-alcoholic beverages , whole grains and plant oil. Moreover, the Authors found in HT patients a significant positive association between plant oil consumption and triiodothyronine levels and a negative association between olive oil consumption and systolic blood pressure in the sub-group of HT patients on levothyroxine therapy (37). 

Line 193-194 ‘The difference between red fresh meat and processed ones is low fats content and absence of additives’.

      is that the only difference? what about the fact that processed meat is classified as a Class 1 carcinogen by the World Health Organization.

      Following the Referee’s suggestion, It has been added in the text: Moreover, processed meat is classified as a Class 1 carcinogen by the World Health Organization

In addition, among the search terms, the authors listed the Mediterranean diet (Materials and Methods), but most of the studies described are only nutrients/product groups whose impact is well known.

No studies in which the effect of the Mediterranean diet (as a pattern/per se) on the thyroid axis/thyroid hormone metabolism was verified. Authors could have at least presented the amounts of the ingredients that were described (I, Se, Fe..) in the Mediterranean diet.

Studies regarding the effect of Mediterranean diet or its components on thyroid axis/thyroid function are discussed in paragraphs 1,4,5 of our paper and related references are cited in bibliography

Moreover, is curcumin a characteristic part of the Mediterranean diet? (Line 334-336).

Even if extracted from the root of an oriental plant, curcumina, as a vegetable with anti-inflammatory and anti-oxidant properties and for its contents in vitamins and minerals, is usually used in Mediterranean diet

Discussion section is disproportionate to the rest of the text

There was a mistake in submitting the paper, paragraph 6 was Conclusions and not Discussion. Actually  the discussion was inserted in each paragraph regarding relationship between Mediterranean diet and thyroid.

We hope that the enclosed paper, revised along the lines indicated, may now be reconsidered for publication in Nutrients